

# MetaBAT 2: an adaptive binning algorithm for robust and efficient genome reconstruction from metagenome assemblies

Dongwan D. Kang[1], Feng Li[2], Edward Kirton[1], Ashleigh Thomas[1], Rob Egan[1], Hong An[2] and Zhong Wang[1,3,4]

[1] Department of Energy, Joint Genome Institute, Walnut Creek, CA, USA
[2] School of Computer Science and Technology, University of Science and Technology of China, Hefei, Anhui, China
[3] Environmental Genomics and System Biology Division, Lawrence Berkeley National Laboratory, Berkeley, CA, USA
[4] School of Natural Sciences, University of California at Merced, Merced, CA, USA

Corresponding author
Zhong Wang, zhongwang@lbl.gov

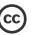

## ABSTRACT

We previously reported on MetaBAT, an automated metagenome binning software tool to reconstruct single genomes from microbial communities for subsequent analyses of uncultivated microbial species. MetaBAT has become one of the most popular binning tools largely due to its computational efficiency and ease of use, especially in binning experiments with a large number of samples and a large assembly. MetaBAT requires users to choose parameters to fine-tune its sensitivity and specificity. If those parameters are not chosen properly, binning accuracy can suffer, especially on assemblies of poor quality. Here, we developed MetaBAT 2 to overcome this problem. MetaBAT 2 uses a new adaptive binning algorithm to eliminate manual parameter tuning. We also performed extensive software engineering optimization to increase both computational and memory efficiency. Comparing MetaBAT 2 to alternative software tools on over 100 real world metagenome assemblies shows superior accuracy and computing speed. Binning a typical metagenome assembly takes only a few minutes on a single commodity workstation. We therefore recommend the community adopts MetaBAT 2 for their metagenome binning experiments. MetaBAT 2 is open source software and available at https://bitbucket.org/berkeleylab/metabat.

# INTRODUCTION

Studies of microbial communities based on microbial isolation and cultivation have been gradually replaced by high throughput, whole genome shotgun sequencing based metagenomics (reviewed in *Van Dijk et al., 2014*; *Tringe & Rubin, 2005*). Advances in computational metagenomics have produced tools that assemble billions of short sequence reads derived from deep metagenome sequencing into larger fragments (contigs), and

subsequently group them into draft genomes by metagenome binning (reviewed in *Kang, Rubin & Wang, 2016*).

Recently we have witnessed exciting progress in metagenome binning as several automatic binning tools have become available. Our group developed MetaBAT (*Kang et al., 2015*) in 2015, among a few others developed around the same time, including MyCC (*Lin & Liao, 2016*), MaxBin 2.0 (*Wu, Simmons & Singer, 2015*), MetaWatt-3.5 (*Strous et al., 2012*) and CONCOCT (*Alneberg et al., 2014*). These binning software tools have achieved various extents of success with simulated data or real world data. However, in practice the quality of binning experiments is largely dependent on characteristics of the underlying dataset and hence the choice of binning parameters. Our users and ourselves both independently observed that MetaBAT's binning performance can vary greatly among different parameter choices. As there are no established parameter optimization methods, to get a comprehensive binning result one has to run multiple binning experiments with different sets of parameters followed by merging the results. For example, in a recent large scale study of over 1,500 metagenome datasets, 8,000 draft genomes were obtained by merging five MetaBAT binning results, each derived from a different parameter set (*Parks et al., 2017*).

In the recent critical assessment of metagenome interpretation (CAMI) metagenome binning challenge (*Sczyrba et al., 2017*), MetaBAT is the fastest and most robust software that can scale up to handle large metagenomic datasets with millions of contigs. Its accuracy was not the best, however, likely due to its lack of consistent binning performance toward various datasets. We therefore replaced the core binning algorithm with a completely new one and report MetaBAT 2 (the original MetaBAT hereafter referred as MetaBAT 1) in this study. The new algorithm consists of several new aspects: (1) normalized tetra-nucleotide frequency (TNF) scores, (2) a graph structure and an iterative graph partitioning procedure for clustering and (3) additional steps to recruit smaller contigs. In addition, we greatly improve the computational efficiency so that the increase in calculations does not affect the program's scalability.

MetaBAT 2 has been packaged by the research community as a Bioconda package (https://bioconda.github.io/recipes/metabat2/README.html) and as a standard APP on the DOE Knowledgebase platform (https://kbase.us/applist/apps/metabat/run_metabat/release). A Docker image is also available (https://hub.docker.com/r/metabat/metabat). There are numerous studies that have reported using MetaBAT 2 and its associated tools for successful large scale metagenomic analyses (*Rinke et al., 2018*; *Bahram et al., 2018*; *Pasolli et al., 2019*). Here, we focus on describing how MetaBAT 2 works, while providing performance benchmarks on a few synthetic datasets and a large number of real world datasets.

## METHODS

### The adaptive binning algorithm

MetaBAT 2 uses the same raw TNF and abundance (ABD) scores as those in MetaBAT 1. There are three major changes in binning algorithms as listed below.

## Score normalization

Integrating TNF score and ABD score is challenging because their distributions can be very different, we therefore applied quantile normalization to TNF score using ABD score distribution. After the normalization, a composite score ($S$) is calculated by taking the geometric mean of TNF score and ABD score. $S$ is subsequently used as the edge weight in the graph-based clustering (below),

$$S = \text{TNF}^{(1-w)} * \text{ABD}^w \tag{1}$$

where $w = \text{nABD}/(\text{nABD} + 1)$, and nABD represents the number of effective samples which have enough coverage (by default >1) for at least one of the contigs. When the number of samples increases, ABD becomes more reliable, so $w$ effectively decreases the relative weight of TNF and increases the weight of ABD. Whenever there are three or more samples available, an ABD correlation score (COR) is also calculated using the Pearson correlation coefficient and then rank-normalized using ABD. In this case, $S$ is calculated as the geometric mean of TNF, ABD, and COR.

$$S = \text{sqrt}(\text{TNF}^{(1-w)} * \text{ABD}^w * \text{COR}) \tag{2}$$

In this way all scores fall within the same range. $S$ should be more accurate than TNF alone when there are multiple samples. In complex communities where TNF score does not distinguish closely related species or strains, $S$ may separate them because it considers coverage covariances.

## Graph-based clustering

Instead of the modified $k$-medoid clustering algorithm implemented in MetaBAT 1, MetaBAT 2 uses a graph based structure for contig clustering. A graph with contigs as nodes and their similarity as edges is constructed in two steps. During the first step, an initial graph is constructed by only using TNF scores. Since TNF scores usually are not very reliable, we only use strong TNF scores for the first stage graph. Here we also put a limit on the number of edges per node to reduce computation, a parameter that can be adjusted to control sensitivity/specificity.

The second step is an iterative procedure of graph building and graph partitioning. At each iteration, a subset of edges with the highest similarity scores ($S$) are incorporated into the above graph, followed by graph partitioning using a modified label propagation algorithm (LPA, (*Zhu & Ghahramani, 2002*)). We modified the LPA in two aspects: (1) we let the search order decided by edge strength so that the partitioning is deterministic rather than random; and (2) we use the previous round of partitioning results as the labels for the graph in the next round to speed up binning. In addition, we used Fisher's method to evaluate the most likely membership of a contig among multiple neighborhoods where the contig is connected to. It is particularly helpful when the sizes of neighborhoods are different. In applying the Fisher's method, we used $1-S$ as $p$-values so that strong scores represent strong probabilities of the connections.

## Small contigs/bins recruiting

MetaBAT 1 by default uses contigs 2.5 kb or larger. As many metagenome assemblies contain smaller contigs, MetaBAT 2 includes an additional step to include small contigs (between 1 and 2.5 kb), as well as the contigs from small bins (<200 kb) if there are three or more samples in the dataset. In this additional step, a "free" contig is assigned to a specific bin where its average correlation to member contigs from that bin is larger than the average correlation among the contigs within the bin.

## Metagenome assemblies used for benchmarking binning

Three synthetic datasets (Low-, Medium- and High-complexity, respectively) were downloaded from the CAMI website (*Sczyrba et al., 2017*).

A total of 120 real world metagenome assemblies were obtained from The Integrated Microbial Genomes & Microbiomes system (IMG/M: https://img.jgi.doe.gov/m/) (*Chen et al., 2018*). A complete list of the samples and their IMG access IDs are available in Table S1. As most of the metagenomics studies today were done on single samples, all of these 120 datasets contain only a single sample. They covered a variety of different environments, therefore providing a realistic estimate of MetaBAT2's performance. It is worth noting that the performance of MetaBAT2 were severely impaired without coverage information from multiple samples.

## Software tools used for benchmarking binning

The other software tools we used are their latest versions: CONCOCT 0.4.0, MaxBin 2.2.4, MyCC (Docker image 990210oliver/mycc.docker:v1), BinSanity v.0.2.6.4 and COCACOLA Python version (updated on March 5, 2017), respectively. All tools were run with their default parameters.

## Searching best parameters by a genetic algorithm

The genetic algorithm was performed using the following parameters: population size: 10, selection size: 3, mutation rate: 0.05, crossover rate: 0.01, minimum/maximum generations: 3/10, and binary tournament selection. To evaluate the performance of binning, we used the minimum information about metagenome-assembled genome standards described in (*Bowers et al., 2017*). The number of high-quality putative genomes was used as the fitness score, where high-quality is defined as $\geq$90% complete and $\leq$5% contamination as determined by CheckM, $\geq$18 tRNAs identified by tRNAscan-SE (*Lowe & Eddy, 1997*), and all three ribosomal subunits, found by cmsearch. While tRNA and rRNA annotations can be annotated just once per contig, CheckM (*Parks et al., 2015*) must be run on each parameter set's results and is the time-limiting step.

## Computational optimization

The above changes in the binning algorithm in MetaBAT 2 require significantly more computation than MetaBAT 1. To make MetaBAT 2 work well with similar computing resources in a comparable runtime, we implemented several computational optimization techniques to improve its resource efficiency.

## Computing efficiency

In addition to the original multi-thread strategy in MetaBAT 1, we also applied a lower level optimization on CPU cache memory access. A typical CPU has only 8–64 kb level-1 cache and 256 kb–2 MB level-2 cache. When the data is bigger than the level-2 cache (e.g., TNF distance matrix), the data is kept in random-access memory that is much slower to access. We adopted a loop tiling threading model that divides the TNF distance matrix into many smaller "tiles" that fit into level-2 cache, and distributes the calculation of each tile among many threads. This optimization alone gains a 35% performance improvement over the original parallel code.

## Memory efficiency

With thousands or even millions of contigs, the pair-wise distance matrix gets bigger and can take a large amount of memory. To avoid storing the entire matrix in RAM, we use a priority queue data structure to store only the top $k$ strong links of every contig for iterative clustering. This method scaled down the memory usage from $O(N^2)$ to $O(N)$, at the cost of a few extra calculations. The parameter $k$ will affect the sensitivity and specificity of binning results, as smaller $k$ values lead to high specificity but low sensitivity.

# RESULTS

## Accuracy benchmarks on synthetic and real world metagenome assemblies

During the first CAMI challenge, MetaBAT 1 was the fastest and most scalable software, but its accuracy was only average compared with CONCOCT, MyCC and MaxBin 2.0 (*Sczyrba et al., 2017*). To test whether MetaBAT 2 improves binning accuracy, we benchmarked MetaBAT 2 by comparing it against MaxBin2, CONCOCT, and MyCC. In addition, we added BinSanity (*Graham, Heidelberg & Tully, 2017*) and COCACOLA (*Lu et al., 2017*), two new automatic binners developed after CAMI 1.0, for comparison. All the tools were run using their default parameters (Methods).

The CAMI metagenome datasets were simulated with different species complexity and genome sizes, consisting of reads from 700 microbial genomes including strain-level diversity and 600 plasmids and viruses (*Sczyrba et al., 2017*). Three datasets were simulated at different complexity levels (high, medium and low complexity). To date, they represent the best benchmark datasets for metagenome binning with known ground truth. We therefore ran the above tools on these three datasets. We used the same accuracy measures as we did in MetaBAT 1, that is, number of genomes recovered at certain genome completeness (recall, 0.5, 0.6, 0.7, 0.8 and 0.9) and certain precision (0.9 and 0.95) cutoffs. The results are shown in Fig. 1.

MetaBAT 2 shows better performance over the other tools in these experiments. In the CAMI Low Complexity dataset, MetaBAT 2 recovers the most genomes at almost every completeness/precision cutoff, except that CONCOCT recovers one more genome at the 90% completeness cutoff. In the other two datasets with higher community complexity, MetaBAT 2 bins more genomes than any other tool tested at every threshold. The difference seems to be more pronounced when complexity increases. For example,

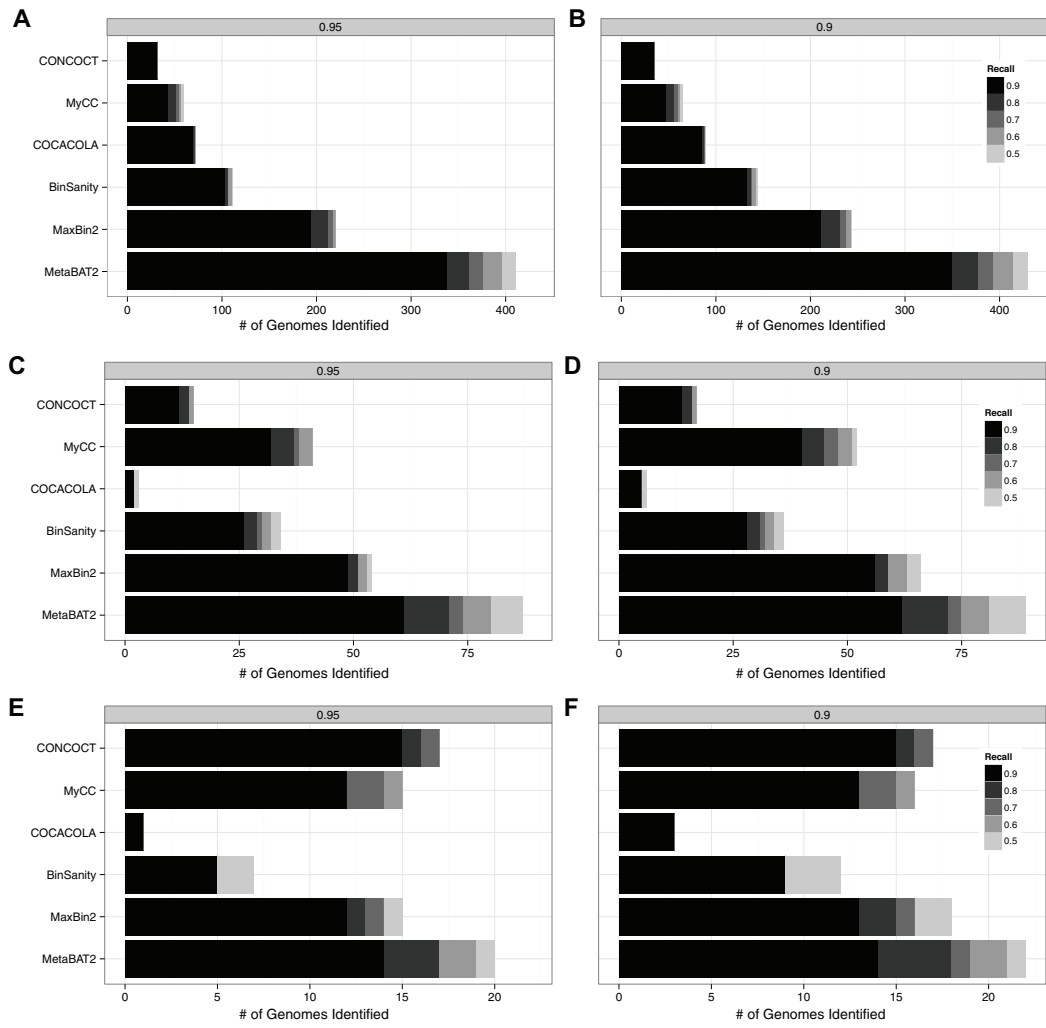

**Figure 1 Benchmark of several popular binning tools on CAMI challenge datasets.** The number of identified genomes are shown at two different precision levels, ≥95% (A, C and D) or ≥90% (B, D and F). The number of identified genomes recovered with a completeness (recall) level 90%, 80%, 70%, 60%, or 50% are represented by different shades of gray, with 90% being the darkest. Benchmarking results using the high complex dataset (A and B), medium complex dataset (C and D), and low complex dataset (E and F) are shown. All the tools (MyCC, CONCOCT, COCACOLA, BinSanity, MaxBin 2 and MetaBAT 2) were run using their default parameters. Completeness and precision were calculated with the ground truth of each dataset.

at 90% completeness and 95% precision levels MetaBAT 2 recovers 333 out of 753 genomes (44.2%) from the CAMI High Complexity dataset, while the next best software, MaxBin2, only recovers 195 genomes (25.9%). These results suggest the adaptive binning algorithm implemented in MetaBAT 2 can adapt to very complex microbial communities.

We also carried out benchmarking experiments on real world metagenome datasets downloaded from IMG/M (*Markowitz et al., 2011*, *2013*; *Chen et al., 2018*). We chose 120 metagenomes assembled from very diverse environmental samples as our test dataset (Methods). All these metagenome datasets were assembled with metaSPAdes (*Nurk et al., 2017*). Hereafter, we refer this dataset as IMG100.
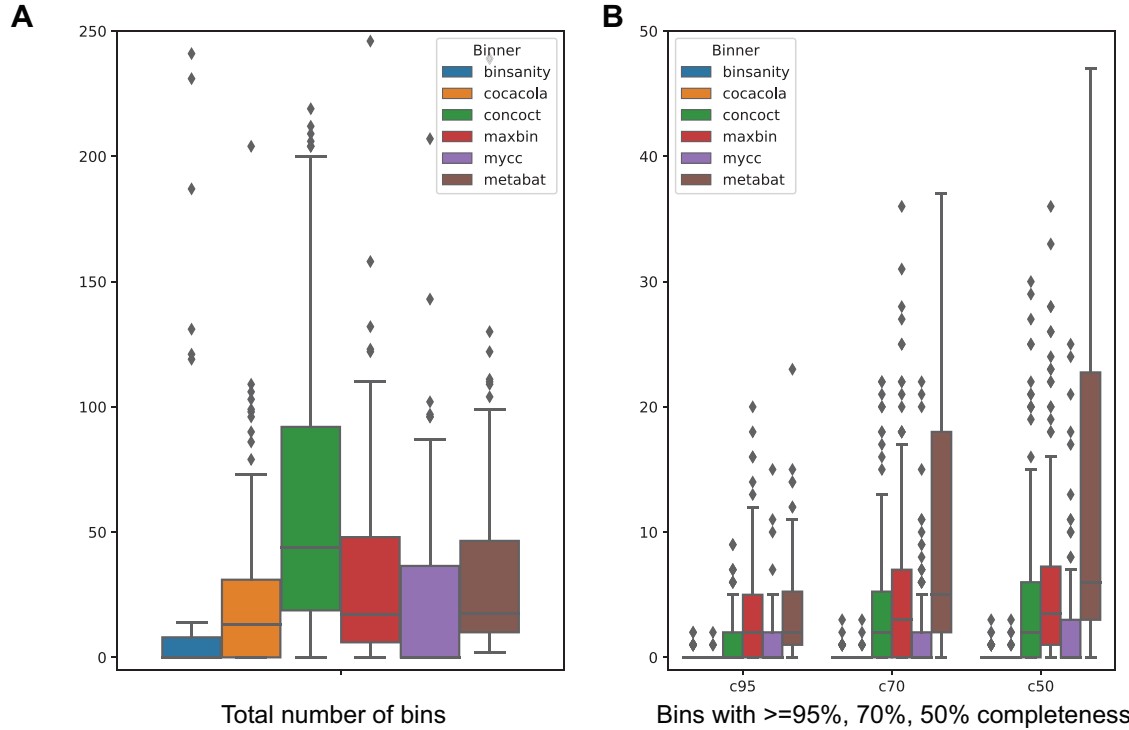

**Figure 2 Comparing binning performance of MetaBAT 2 with alternative binning tools on real world metagenomes.** A total of 120 meta-genome assemblies were obtained from IMG (IMG100, see Methods). (A) Total number of bins from each of the dataset formed by each binner. (B) Using a 5% contamination cutoff, number of genome bins that have at least 95% (c95), 70% (c70) and 50% (c50) genome completeness estimated by CheckM. Experiments that produced no bins were omitted.

Most of these metagenome datasets produced very few bins from all software tools. Some datasets failed to produce more than five bins from any of the tools ($n = 11$), and binning for some datasets failed or did not finish for all the tools in 72 h wall time ($n = 36$, Table S3). Figure 2 shows the performance of each method on IMG100 metagenomes. Overall MetaBAT2 produced a moderate number of total genome bins. But when compared the number of pure genome bins ($\geq$95%), for those fulfilling different completeness criteria of 95%, 70%, and 50%, MetaBAT 2 consistently outperforms other tools.

## The default set of parameters is good for most datasets

We next ask the question whether or not MetaBAT 2 requires different parameter sets for optimized accuracy for different datasets. There are only three parameters that may affect binning accuracy: (1) maxEdges (the maximum number of edges a node can have when constructing the graph, a lower number should reduce computing time but may also reduce sensitivity); (2) maxP (percentage of high quality contigs included for binning, a higher number gives more sensitivity); and (3) minS (the minimum score of an edge kept for binning, a higher number gives more specificity).

We designed a genetic algorithm to attempt to optimize the above parameters using the IMG100 dataset (Methods). For each of the samples in this dataset, the genetic algorithm

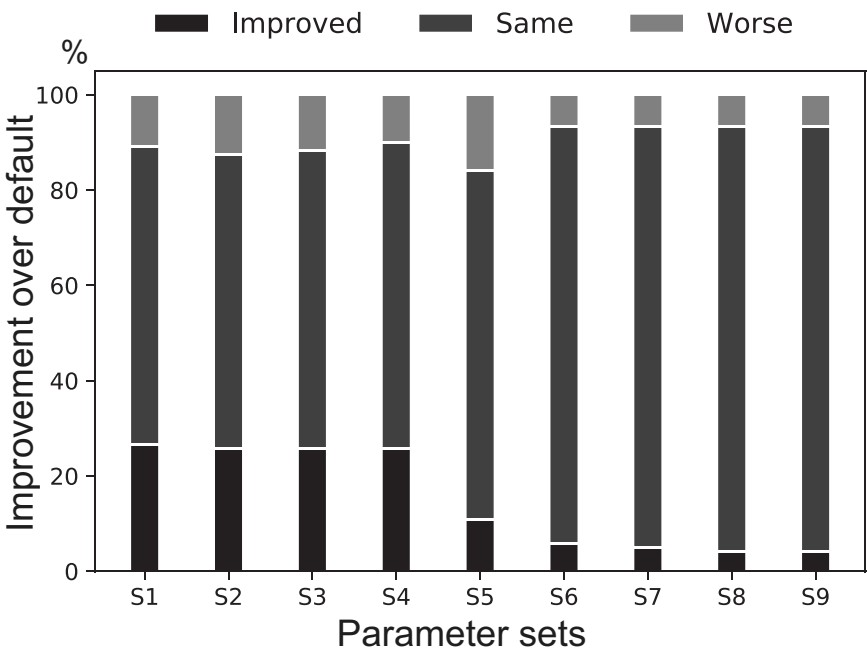

**Figure 3 A binning performance comparison between the default parameter set of MetaBAT 2 against several common best parameter sets found by the genetic algorithm.** The IMG100 dataset was used for searching for the best parameter set for each sample. For each parameter set (S1–S9, see Table S2), a stacked bar shows the percentages of datasets where its performance is better than (darkest gray at the bottom), the same as (medium gray in the middle), or worse than (light gray at the top) the default parameter set. Overall the default parameter set is consistently selected as the best parameter set for most samples.

systematically explores the parameter space and attempts to find the best parameter set for this sample. In this experiment we only considered the high-quality genome bins for performance scoring (Methods). Among all the best parameter sets found from all samples, the default parameter set is selected to be the best for the majority (58%) of the samples (Table S2). Comparing the next nine most frequently selected best parameter sets to the default one on all the samples showed that the default parameter set has a consistent performance (Fig. 3). The genetic algorithm did find some parameter sets that are slightly better than the default on some samples under this scoring metric, although it significantly increased the total running time. Using a different scoring scheme or using a different set of testing data may select a different set of parameters, but the benefit over the default parameters appears to be very small (Fig. 3).

We also experimented with MetaBAT 1 on the IMG100 set, using the two most commonly used preset parameters (sensitive and superspecific). In this comparison setting we can see whether or not MetaBAT 2 with default parameters is more sensitive than MetaBAT 1 (sensitive) and more specific than MetaBAT 1 (very specific). Figure 4 shows the performance of each method in the top 20 metagenomes ordered by the number of genome bins identified. In the majority of the binning experiments, MetaBAT 2 outperforms both modes of MetaBAT 1, demonstrating a robust performance without parameter tuning.

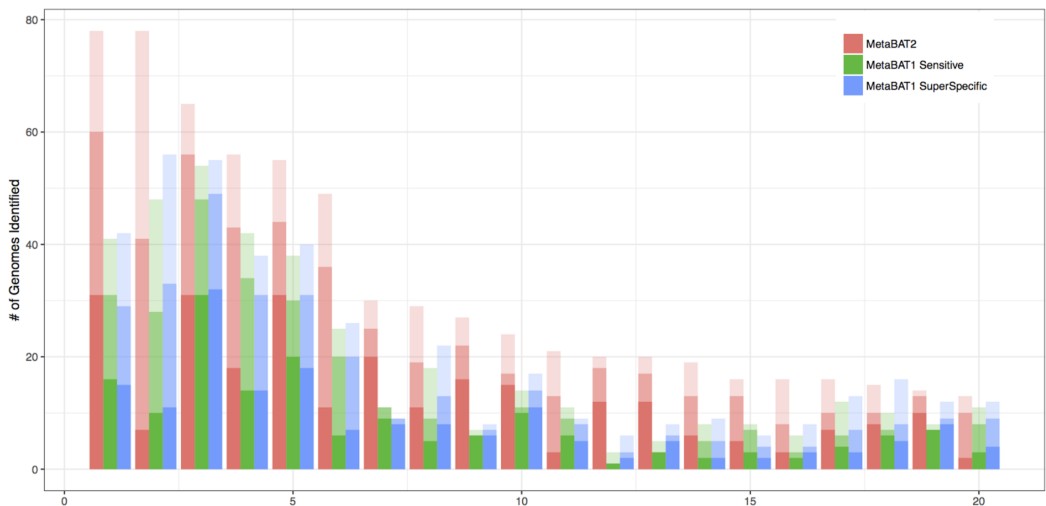

**Figure 4 Comparing MetaBAT 2 with two sets of MetaBAT 1 binning experiments using real world metagenomes.** IMG100 dataset was used for benchmarking experiment. The top 20 metagenomes ordered by the number of genome bins identified are shown. X-axis represents each metagenome, and Y-axis shows the number of genome bins identified using 5% contamination cutoff. Each bar represents three completeness results of 90%, 70%, and 50% by the order of color density (i.e., darkest color represents 90%). The completeness and contamination were estimated by CheckM. MetaBAT 2 outperforms both modes of MetaBAT 1 in most cases.

**Table 1 Runtime and memory comparison on CAMI high, medium, low dataset.** All tests were run on a workstation with two CPUs of Intel(R) Xeon(R) CPU E5-2698 v3 @ 2.30 GHz, 128 GB Memory.

| Runtime memory | MetaBAT 2 | MetaBAT 1 | CONCOCT | MaxBin | MyCC | BinSanity | COCACOLA |
|---|---|---|---|---|---|---|---|
| CAMI high | 1 min 54 s | 16 min 53 s | 2 h 40 min | 7 h 1 min | 6 h 5 min | 5 h 42 min | 14 h 2 min |
| | 2.63 GB | 3.02 GB | 1.28 GB | 2.99 GB | 3.04 GB | 50.82 GB | 4.95 GB |
| CAMI medium | 25 s | 3 min 15 s | 1 h 21 min | 1 h 10 min | 1 h 26 min | 2 h 3 min | 3 h 1 min |
| | 0.56 GB | 0.78 GB | 0.96 GB | 1.04 GB | 2.10 GB | 25.52 GB | 1.21 GB |
| CAMI low | 7 s | 47 s | 13 min 52 s | 11 min 5 s | 20 min 25 s | 18 min 47 s | 30 min 8 s |
| | 0.16 GB | 0.16 GB | 0.38 GB | 0.69 GB | 0.82 GB | 2.73 GB | 0.85 GB |

## Benchmarking MetaBAT 2's computing efficiency

In contrast to most of the other software tools that are implemented with Python, MetaBAT 2 is written in C++ with extensive low-level computational optimization (Methods). This gives MetaBAT 2 a unique advantage in computational efficiency. The runtime and memory consumption of MetaBAT 2 and alternative tools on CAMI High, Medium, and Low complexity datasets are shown in Table 1. The tests were run on a workstation with 2 Intel Xeon CPUs @ 2.30GHz, each with 16 cores and 40MB smart cache, and 128 GB RAM.

MetaBAT 2 finished the CAMI Low dataset in just 7 s, while most other tools took 11 min or more, which is 90 times or more slower than MetaBAT 2. It finished the Medium and the High datasets in 25 s and 1 min 54 s, respectively. The others need from 1 h to several hours for binning the two datasets. Memory requirement by all tools varies.

In general, MetaBAT 2 requires the least while BinSanity requires the most (with a factor of 20×, in contrast to others with about 1–4×).

Running on the entire IMG100 datasets we observed a much more pronounced difference in computational resource requirement. MetaBAT 2 finished binning in a few hours, but other tools took days even weeks, if they can finish. Some tools failed on some of the large assemblies after a long time probably due to their high memory requirement.

## DISCUSSION

There are a couple of considerations when using MetaBAT 2. First of all, MetaBAT 2 uses an adaptive binning algorithm which puts more weight on ABD but less weight on TNF, which makes it work well on datasets with multiple samples (CAMI synthetic sets, e.g.). In general we expect more samples to produce better accuracy. For single-sample datasets, some users reported a genome can sometimes be split among different bins in spite of consistent TNF composition. This illustrates that MetaBAT 2 weighs heavily on purity with some sacrifice in completeness. A manual, post-binning polishing step may be required to further improve completeness. A second consideration is that MetaBAT does not eliminate chimeric contigs or other artifacts from assembly, so binning results from very poor assemblies will not be reliable. Some post-binning polishing process, such as $d_2^S$ Bin (*Wang et al., 2017*), may help reduce the contamination problem.

The method of normalizing TNF and CORs using ABD makes it possible for future MetaBAT versions to incorporate additional scoring matrices in a similar fashion. Interestingly, in a recent preprint, *Nissen et al. (2018)* developed an alternative strategy based on deep variational autoencoders to accomplish a similar task and got very good results. Additional matrices could be taxonomic similarity, physical linkage (provided by Hi-C experiments or paired end reads), and/or other similarity matrices of the contigs. Incorporating taxonomic information would provide a framework to unite taxonomy dependent and independent binning strategies.

## CONCLUSIONS

In conclusion, we show that the adoption of a new adaptive binning algorithm makes MetaBAT 2 automatically adapt to datasets with various characteristics and provides robust metagenome binning. This should greatly reduce users' time needed to manually explore the underlying datasets and experiment with different parameter sets. This capability should be particularly useful for datasets derived from unknown complex microbial communities, as empirically setting parameters might be challenging. Extensive low-level computational optimization taking advantage of the underlying hardware capabilities also makes MetaBAT 2 run very efficiently, and makes it scalable for very large datasets.

### Funding

Dongwan Kang, Edward Kirton, Ashleigh Thomas, Rob Egan, and Zhong Wang's work was supported by the U.S. Department of Energy, Office of Science, Office of Biological

and Environmental Research under Contract No. DE-AC02-05CH11231. Feng Li was supported by an exchange student fellowship from the China Scholarship Council (CSC). The funders had no role in study design, data collection and analysis, decision to publish, or preparation of the manuscript.

### Grant Disclosures

The following grant information was disclosed by the authors:
U.S. Department of Energy, Office of Science, Office of Biological and Environmental Research: DE-AC02-05CH11231.
China Scholarship Council (CSC).

### Competing Interests

The authors declare that they have no competing interests.

### Author Contributions

- Dongwan D. Kang conceived and designed the experiments, performed the experiments, analyzed the data, prepared figures and/or tables, authored or reviewed drafts of the paper, approved the final draft.
- Feng Li performed the experiments, analyzed the data, prepared figures and/or tables, authored or reviewed drafts of the paper, approved the final draft.
- Edward Kirton performed the experiments, analyzed the data, approved the final draft.
- Ashleigh Thomas performed the experiments, prepared figures and/or tables, approved the final draft.
- Rob Egan conceived and designed the experiments, approved the final draft.
- Hong An authored or reviewed drafts of the paper, approved the final draft.
- Zhong Wang conceived and designed the experiments, authored or reviewed drafts of the paper, approved the final draft.

### Data Availability

MetaBAT 2 is available at https://bitbucket.org/berkeleylab/metabat.

### Supplemental Information

Supplemental information for this article can be found online at http://dx.doi.org/10.7717/peerj.7359#supplemental-information.

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
