# Peer review of "MetaBAT 2: an adaptive binning algorithm for robust and efficient genome reconstruction from metagenome assemblies"

_PeerJ, doi:10.7717/peerj.7359_

## Round 0.1 · original submission · Major Revisions

Dear Dr. Kang and colleagues:

Thanks for submitting your manuscript to PeerJ. I have now received two independent reviews of your work, and as you will see, the reviewers raised some concerns about the manuscript. Despite this, these reviewers are optimistic about your work and the potential impact it will have on research communities studying processing of metagenomic data. Thus, I encourage you to revise your manuscript accordingly, taking into account all of the concerns raised by both reviewers.

In your revision, please be transparent on the methods for comparative analyses, and also improve the overall clarity of your methods. For instance, provide the shell commands and post-processing scripts, and all information necessary such that your work can be repeated.

Please note: while most of the concerns of the reviewers are relatively minor, this is a major revision to ensure that the original reviewers have a chance to evaluate your responses to their concerns.

I look forward to seeing your revision, and thanks again for submitting your work to PeerJ.

Good luck with your revision,

-joe

·

Basic reporting

The article "MetaBAT 2: an adaptive binning algorithm for robust and efficient genome reconstruction from metagenome assemblies" describes an algorithm for metagenome binning. The main purpose is to describe the improvements made to the algorithm previously published as MetaBAT and to evaluate the performance of this algorithm on three artificial data sets and a large number of real metagenomes.

The article is well written with a language that was easy to understand. I also find the structure and presentation to be well organized. However, the following suggestions may improve the language even further.

Line specific language comments:
Line 38: "tools become available" -> "tools have become available"

Line 50 states: "MetaBAT is the … most robust software" but on line 52: "due to its lack of robustness", this seems slightly contradictive to me. Please, adjust it.

Line 50: "can scale up to large" -> "can scale up to handle large"

Line 54: Please expand the abbreviation 'TNF' as this is the first time it is mentioned

Line 60: I believe "docker" should be "Docker" as it is a company name

Line 70: Line numbering seems broken here

Line 70: "where nABD represents" -> "and where nABD represents"

Line 83: "The LPA was modified so that the partitioning is deterministic since the search order is decided by edge strength". Check grammar: "since" does not fit perfectly.

Line 158: "from a very diverse environmental sample" -> "from very diverse environmental samples" or "from a very diverse set of environmental samples"?



The authors also show a great understanding of the field and that they are up to date with the latest advances. However, I find the article lacking in available raw data. Especially in the form of code used to perform the comparisons and analysis throughout the article, but also the results behind figures 2 and 4 where only a subset of the samples are shown should be made available as supplementary tables. Regarding the code, while the code for the MetaBAT2 tool itself is supplied as a bitbucket link (unfortunately without a doi, commit hash or detailed version number), I cannot find any code corresponding to how the different tools were ran.

Specifically I'd like to see:
Scripts or commands showing how the data sets were preprocessed for the different tools
Scripts or commands showing how the different tools were run
This should be supplied for both the CAMI datasets and for the IMG100 dataset.

Furthermore, these scripts should make it clear whether the IMG100 samples have been binned separately, without coverage information from additional samples. If this is the case, this should also be clarified and discussed in the main text. This is especially important since binning of individual samples is quite a different task than binning using the information from multiple samples.

Additional comments:
Line 100: The versions are stated to be the "latest". Please add a date for when this check was done. At least for some of the programs (e.g. CONCOCT), more recent versions are already available.

Line 102: "All tools were run with their default parameters" - This comment is not a sufficient description on how the comparisons were made. Preferably, complete scripts to reproduce the comparison should be supplied. Especially, the commands for input preprocessing is important as this should differ between the tools.

Experimental design

MetaBAT2 is arguably the most used metagenome binning algorithm of today. To have a detailed description of the internals of this algorithm is of great value for the research community. This article sets out to fill that knowledge gap and goes a long way in doing so. However, some details need further clarification regarding the algorithm, specifically:

First part of "Line 70": "We use ABD to rank-normalize TNF (where the smallest TNF becomes the smallest ABD and the greatest TNF becomes the greatest ABD, and so on)" - I cannot understand what this means in its current form. I assume TNF and ABD means “TNF score” and “ABD score”. Although, with this transformation, does that mean TNF scores are replaced by ABD rank? Then it stops being a TNF score, right? Please expand on the concept of rank-normalization and what purpose it serves in the algorithm.

Line 70, the definition of w: My interpretation of this is that it will give more weight to the ABD score the more samples with sufficient coverage there is. Is this correct? In that case it might be worth pointing out to the reader.

Line 71: "S should be more accurate" - More than what? Why? Please clarify.
Line 71 - 72: "especially when the communities are extremely complex". Why? Please clarify.

Line 76: Please clarify if TNF in "constructed by only using TNF" is TNF score or the actual TNF.

Line 81: Are edges discarded if a node has reached it’s limit? How large is the subset per iteration?

Line 85: I was not able to find an explicit mentioning of Fisher’s method in the MetaBAT 1 paper. Please explain how this is used in MetaBAT 2.

Line 91: "correlation to member contigs from that bin is larger" Please clarify if this is intends the average correlation for all contigs in the bin or if just the maximum correlation value is intended?

Line 104: A more explicit description of "The genetic algorithm" would be beneficial. Currently only the parameters are listed.

Furthermore, as I've mentioned in above, scripts or commands on how the investigation was performed needs to be submitted. This is essential for the reproducibility of the results and should definitely be included as supplementary data. In the manuscripts current form, it is not clear how parts of the comparison was performed.

Furthermore, it seems to me (not entirely clear due to above mentioned reason) that the comparison on real metagenomes are not using coverage over multiple samples. The possibility to use coverage over multiple samples is what distinguishes the metagenomic binning tools released at around 2015 from the previous generation of binning softwares. To compare these tools without using that possibility (especially without explicitly commenting on it) seems to me slightly odd. Please, clarify why the real dataset was chosen so that multiple samples were not possible to use. Or if my understanding is in fact wrong, please make this clearer in the text.

Validity of the findings

MetaBAT2 is most likely the fastest binning algorithm available today, and the authors make that statement clear with the data included in the results section. I also greatly appreciate the good description on how this computational efficiency is achieved.

The quality of the results from MetaBAT2 is also most likely very good although I am not completely convinced that the evaluations included in the manuscript shows this in a fair way. I might have misunderstood the following, and in that case, please clarify in the manuscript so that this misconception is avoided. For example, it appears as Figure 2 includes a subset of samples where specifically MetaBAT2 was most successful. The result from the other samples (~90%) are not shown and not included in any supplementary data. Also, comparing the results of the competitors in Figure 1 to Figure 2, panel b in Szcyrba et al. 2017, all three competitors which are included in both comparisons (MaxBin2, COCACOLA and CONCOCT) show poorer performance in this manuscript than in the previously published article. Part of this might be due to a parameter optimization performed in the Szcyrba publication which is not reflected in the default settings of these programs. The settings used for Szcyrba et al. should be readily available from the corresponding biocontainers and I would argue that that would be a more fair comparison. The performance displayed by MetaBAT2 in this manuscript would still place it in the first place using the results from Szcyrba et al. 2017, but the competition would be more close.

Line specific comments:
Line 162: "only producing fewer than 5 genome bins" - for MetaBAT2 or for all softwares combined?

Line 163: "the top 13 metagenomes ordered by the number of genome bins identified" - for MetaBAT2 or for all softwares combined? From looking at of Figure 2, the different samples seem to be sorted based on the number of bins from MetaBAT2 only. If this is the case for how the samples shown were chosen as well, this is unacceptable. Choosing the samples where MetaBAT2 performed the best obviously bias the comparison in MetaBAT2:s favour. Please clarify how the 13 samples were chosen based on number of bins from all binning softwares or only from MetaBAT2.

Figure 1: "All the tools … were run using the default parameters": Again, I would argue that this is not enough. Especially since comparing these results to figure 2, panel b of (Sczyrba et al. 2017), all overlapping software (except metabat) performs worse in your comparison than in the already published comparison. In Sczyrba et al., all tools were submitted in reusable, reproducible containers. For this reason, it is very important that the authors of thus manuscript can show how the tools were run in their comparison to ensure that no mistakes have been made.

Line 204-207: Please clarify which "some tools failed on some large assemblies". Perhaps in a supplementary table showing the number of genome bins for all software and all samples?

Additional comments

While not necessary for publication in PeerJ, I feel this paper is of great value to the metagenomic binning research community. However, I find several points that need to be revised before it is fit for publication. While these changes are of great importance, I do not think they will be very difficult to make. To me, the most important point to improve is the transparency on how the comparisons were made and how the results of those are displayed. The second most important point is to expand and clarify the methods section to make it easier for readers to understand the algorithm.

·

Basic reporting

The manuscript is very minimalist in its descriptions of metaBAT2's algorithm. However, this is mostly sufficient considering the main algorithm is covered in detail in the original metaBAT1 publication.

It is important for the authors to provide all the shell commands and post-processing scripts used in this analysis. I recognize the simplicity for some of the benchmarks, however it is still crucial for re-reproducibility purposes.

A minor suggestion for the readability of Figure 2: visually separate the bars coming from different metagenomes to make the segments more distinct.

Experimental design

As expected, the performance benchmarks display the superior performance of metaBAT2 over other software. However, considering that this manuscript is meant to outline the improvements of metaBAT2 over metaBAT1, it is strange that performance of metaBAT1 itself is not included in most benchmarks (Figures 1 and 2, Table 1). Even if the differences are minor, it is important to include this information.

The authors also talk extensively about the scalability of metaBAT2 compared to metaBAT1. In my experience this is accurate, however it it would be nice to see this quantified. Beyond adding metaBAT1 to Table 1, it would be interesting to see a curve displaying the size of a data set vs either total compute time or maximum memory usage.

Validity of the findings

It is interesting to see the very sub-par performance of CONCOCT in the benchmarking of CAMI High, considering its relative success in the original CAMI challenge publication. I wonder if the bench-marking was done without breaking up the longer contigs into smaller sub-sequences, as was intended by the software. See discussion at https://github.com/bxlab/metaWRAP/issues/76.

The metaBAT2 software has already being applied in a variety of microbial metagenomic studies, and it is well known in the research community that currently it delivers unparalleled binning performance in most data. Compared to metaBAT1, which was also popular, metaBAT2 offers significant improvements in performance and computation efficiency in larger data. With this in mind, it is good to see that the manuscript has been written, especially considering the software has been available to the community for some time. However, the article and its conclusions could be greatly improved by a several added benchmarks that would put it into context with the already published metaBAT1 software.

---

## Round 0.2 · Minor Revisions

Dear Dr. Kang and colleagues:

Thanks for revising your manuscript. The sole reviewer is very satisfied with your revision (as am I). Great! However, there is one final concern raised by the reviewer. Please address this ASAP so we may move towards acceptance of your work.

Best,

-joe

·

Basic reporting

The manuscript has greatly improved from the previous version, and sufficient retails are provided.

Experimental design

Thanks to suggestions from Reviewer 1, the methods are more detailed and the authors provided scripts for reproducibility online.

Validity of the findings

Most comments have been addressed, and I thank the authors for providing comparison between the performances of metabat1 and metabat2 to clearly outline what was gained by producing this new version.

One request that was not addressed was the inclusion of metabat1 run-time statistics in Table 1. Considering speed is one of the major improvements of metabat2 over metabat1, I feel it is important to include this comparison. Running metabat1 on default settings is probably sufficient for this.

---

## Round 0.3 · accepted · Accept

Dear Dr. Kang and colleagues:

Thanks for revising your manuscript to PeerJ, and for addressing the concerns raised by the reviewers. I now believe that your manuscript is suitable for publication. Congratulations! I look forward to seeing this work in print, and I anticipate it being an important resource for research communities studying metagenomic data processing.

Thanks again for choosing PeerJ to publish such important work.

-joe